# Urban Green Space Planning, Policy Implementation, and Challenges: The Case of Addis Ababa

**Shibire Bekele Eshetu** [1,*] **, Kumelachew Yeshitela** [2] **and Stefan Sieber** [1,3]

1   Sustainable Land Use in Developing Countries, Leibniz Centre for Agricultural Landscape Research (ZALF), 15374 Müncheberg, Germany; stefan.sieber@zalf.de
2   Ethiopian Institute of Architecture, Building Construction (EiABC), Addis Ababa University, Addis Ababa 1000, Ethiopia; kumelachew.yeshitela@eiabc.edu.et
3   Department of Agricultural Economics, Humboldt University of Berlin, 10099 Berlin, Germany
*   Correspondence: shibire-bekele.eshetu@zalf.de

**Abstract:** Urban forestry and green spaces have less priority in urban planning. This research intends to assess the policy and planning of urban green spaces with their potential implementation status and challenges in planning and implementation. The general objective is to assess urban green space planning, policy, and implementation strategies and challenges encountered in Addis Ababa. The primary data was collected through key informant interviews, focus group discussions, and field observation; secondary data from a literature review along with examining policy and masterplans of Addis Ababa has been used. The mapping of stakeholder and institutional arrangements is analyzed using stakeholders' consultation. Triangulation is used for data validation and analysis. Existing policy and proclamations must be supported by legislative regulations and implementation frameworks that provide the basis for concrete action plans. The incentives stipulated by the forest policy are not implemented to the required level. The 10th masterplan of the city (2017–2027) shows that the city will increase its green area development and public recreation coverage to 30% by 2020. Principles, such as multi-functionality, connectivity, green-grey interaction, and social inclusiveness, are considered in the planning of the green space development in the 10th masterplan. The research concludes that regulations and directives are not clearly drafted by responsible bodies, and low enforcement is hardly applied with respect to the green space development of Addis Ababa.

**Keywords:** green infrastructure; planning principles; strategies; masterplan

## 1. Introduction

The 2030 Agenda for Sustainable Development Goal 11 aims to make cities and human settlements inclusive, safe, resilient, and sustainable [1]. Globally, policymakers are addressing issues of urbanization and unplanned urban growth [2,3]. It is found that urban planning fails in practice in some developing countries because planning is frequently overambitious, especially in light of the capabilities of the administrative system to enforce their implementation [3,4]. Greenspaces in Africa's urban regions are still hardly recognized in policymaking [5].

Greenspace allocation is a common problem across urban centers: it must be accessible by all social groups without segregation [6]. It is shown that there is also a lack of appropriate green infrastructure approaches that integrate well into the planning and governance system of cities [7,8]. To achieve the development of green spaces that provide environmental, economic, and social benefits, the basic principle of connectivity, multi-functionality, and social inclusiveness must be considered during planning [9]. Green spaces should be treated well among the top priorities of the development agenda of urban planning authorities with the allied institutions managing greenspaces [10].

The weak legal and regulatory framework in Sub-Saharan Africa is readily visible, leading to inadequate green space development and management in urban areas [11]. In

the Ethiopian context, although there are policy and legal documents regarding urban green components, quantitative guidelines and standards do not exist [9]. Legislations, guidelines, and standards have an important influence on the planning and implementation of green space components [11,12]. The current masterplan of Addis Ababa provides the framework to organize the city's spaces in an economically productive and environmentally healthy way. The ultimate goal of the plan is to ensure that the city contributes its share in bringing the nation's economy to the level of middle-income countries; in the process, it will also improve the living standard of the city's residents. Despite this, the majority of the developments carried out in the city still do not comply with the masterplan framework [12]. Addis Ababa is selected as a case study site because most development works and infrastructures are intensively implemented in this city. Being the capital of the country, Addis Ababa has gone through different phases of development in terms of grey infrastructure, whereby in some cases compromised the green infrastructure of the city. It is also the only city in Ethiopia that both the Federal and the City administration take role in the area of green space policy and implementation at the same time.

Although there are plenty of challenges that hamper the provision of well-designed urban green spaces in Addis Ababa, there are also potential opportunities to enhance the environmental, economic, and aesthetic value of the city. To realize these green space values, urban areas need proper long-term planning with the execution of projects integrated into the structural plan of the city and the preparation of local development plans prior to the execution of projects. This research evaluates the urban green space planning, policy, and implementation strategies and challenges encountered during implementation processes in Addis Ababa. Specifically, the research seeks to achieve the following objectives: (i) assess urban green space policies in planning and the implementation of green space development in Addis Ababa; (ii) assess institutional arrangements of urban green space planning and implementation in Addis Ababa; (iii) assess challenges and positive initiatives of green space implementation in Addis Ababa; and (iv) assess the principles of urban green space planning in the incorporation of Addis Ababa's masterplan and implementation of the planning principles on the ground.

## 2. Materials and Methods

### 2.1. Study Area

Addis Ababa, the capital city of Ethiopia, is located 9°0′19.4436″ North and 38°45′48.9996″ East in the central part of Ethiopia. Elevation ranges from 3200 m a.s.l at the highest peak of Entoto to 2200 m.a.s.l. at the lower southern part of Akaki plains. Addis Ababa, comprising 520 km$^2$, is the largest city in the country and the biggest city in the world located in a land-locked country without access to the sea [13]. According to the 2007 census, the total population of the city was 3,384,569. Looking at population growth, it is estimated that in 2019 approximately 7,823,600 called the city home, resulting in a population density of 5165 individuals per square kilometer [14]. Figure 1 shows Addis Ababa's green space distribution, plantation forest in the Northern part, urban agriculture in the Southern part, parks distributed in sub-cities and riverside vegetation are the most common green space types distributed in the city.

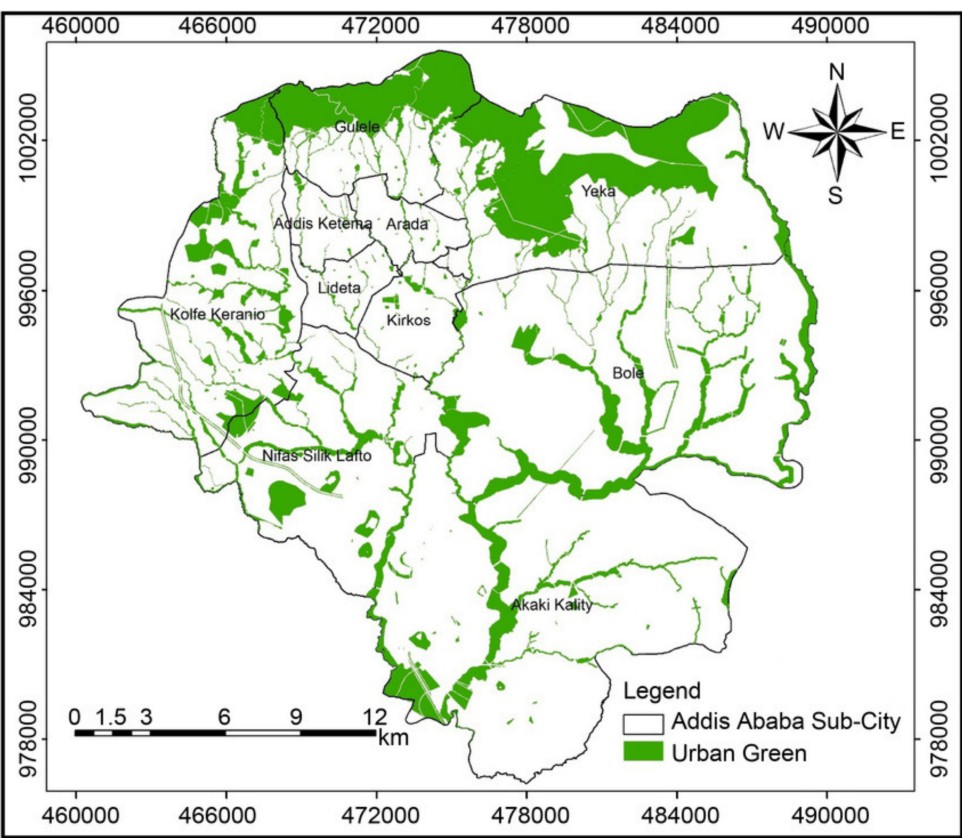

**Figure 1.** Map of Addis Ababa with existing green spaces. Reprinted from ref [15].

Historically, Addis Ababa was covered with natural indigenous vegetation that belongs to Afro-montane forests and woodland. At a high altitude of over 3000 m.a.s.l., the natural forest was dominated by indigenous tree species such as Juniperus procera, Olea europaea subsp cuspidata, Podocarpus falcatus, Hygnia abysinica, and Erica arborea. At the turn of the 20th Century, woody vegetation was removed as the city expanded and the population grew [16]. Woody vegetation was exploited by city dwellers and surrounding farmers for fuel wood consumption and construction. Currently, only a few patches of natural forest remain, while exotic species, especially Eucalyptus, have taken over. Within the city, green space is limited, with few urban parks, some riverside vegetation, and street trees; in the outskirts of the city, there is also a peri-urban forest [17].

*2.2. Data Type and Source*

Both primary and secondary data were collected to undertake this research. The primary data were collected through key informant interviews, a focus group discussion, and field observation. Secondary data includes Google maps data, satellite images, along with a literature review examining policy and masterplans of Addis Ababa. Structured (questions which are developed prior to the interview) and semi-structured questions (questions which were open and developed or reframed during a discussion) were developed to answer the research questions regarding the institutional and governance structures responsible for the green space development of the city. From each institution, two key informants were selected: one expert and the other holding an administrative duty. The key informants are selected from the different government organizations listed in Table 1. Representatives from these institutions participated in the focus group discussion, which was designed to clarify issues raised by the key informants during the individual interview.

**Table 1.** List of organizations approached for key informant interview.

| No | Organization | Roles (Policy, Implementation, Support) |
|----|--------------|------------------------------------------|
| 1 | Ethiopian Environment, Forestry and Climate Change Commission (EFCCC) | Policy draft, regulatory body at the Federal level. |
| 2 | Addis Ababa City Government Environmental Protection and Green Development Commission | Regulatory organization at the city level. |
| 3 | Addis Ababa City Government Plan and Development Commission | Commissioned to do the structural planning of the city, monitoring, and evaluating according to local development plan of the city. Implements green space plans developed by the planning commission. |
| 4 | Addis Ababa City Government Farmers and Urban Agriculture Development Commission | Execution of urban agriculture plan developed by plan and implemented together with departments under sub-cities. |
| 5 | Addis Ababa River Basin and Green Area Development and Administration Agency | Implementation of policies drafted by Addis Ababa City Government Environmental Protection and Green Development Commission and support (cascades to Sub City level). |

Key informant interviews from individuals representing institutions dealing with urban green space, both at the federal level and the City of Addis Ababa, were approached during data collection. The institutions were selected via purposive sampling design, specifically seeking representatives of regulatory bodies and implementers of green space plans in the city. The key informant interviews started with the Ethiopian Ministry of Environment, Forestry, and Climate Change, then, following the snowball sampling technique, as adopted from DenBiggelaar [18], subsequent institutions were selected. The respective selected institutions assigned an expert to participate in the interview as a key informant representing the institution. A total of 10 key informants participated from the 5 institutes, whereby each institute assigned 2 key informants: one expert and one with an administrative role. Questions focused on the mandate of the organization on green space development; if it is more of a regulatory body or an implementer of green space developments. Information, such as policies and guidelines, developed by the organization if any, green space developments that have been implemented by the organization if any, and the challenges and opportunities the city is facing regarding green space development were also covered in the questions.

During the interviews, both administrative and expert views were included. Each interview started with a semi-structured question guide that included space for discussion of related issues as raised by the respondents. The focus group discussion was held after all key informant interviews. Due to the COVID-19 pandemic, the focus group discussion was held in a small group (3 people) from each representative institution listed in Table 1. A total of 5 Focus Group Discussions (FGD) were held. In the FGDs, issues related to institutional mandates and accountability were clarified. The FGDs have been formed considering one key informant is taking part in each group. Policy drafts, masterplan of the city, and other important documents were reviewed from published sources, institute archives, and drafts acquired from the organizations that participated in the key informant interview has been presented in Table 2.

**Table 2.** Reviewed documents dealing with urban green space policy in Ethiopia and Addis Ababa.

| No | Documents Reviewed | Source of Document (Organization/Available at) |
|---|---|---|
| 1 | Urban Planning Proclamation no 547/2008 | https://www.ethioconstruction.net/sites/default/files/Law/Files/Urban%20Planning%20Proclamation%20No.%20574-2008.pdf accessed on 21 May 2020 |
| 2 | Structural plan of Addis Ababa (2002–2012) | Plan commission |
| 3 | Structural plan of Addis Ababa (2017–2027) | Plan commission; https://c40-production-images.s3.amazonaws.com/other_uploads/images/2036_Addis_Ababa_Structural_Plan_2017_to_2027.original.pdf?1544193458 accessed on 14 June 2020 |
| 4 | Forest Regulation of Ethiopia (Final draft prior to approval) | EFCCC |
| 5 | Ethiopia's Climate Resilient Green Economy | Federal Democratic Republic of Ethiopia, Climate Resilient Green Economy Strategy, 2011 by UNDP |
| 6 | The Urban Greenery and Beautification strategy | The Urban Greenery and Beautification strategy, 2015, River Basin and Green Area Development Administration Agency. |
| 7 | Environmental Policy of Ethiopia | http://extwprlegs1.fao.org/docs/pdf/eth133155.pdf accessed on 14 May 2020 |
| 8 | Environmental Protection Organs Establishment proclamation, Proclamation No. 295/2002 | http://extwprlegs1.fao.org/docs/pdf/eth44280.pdf accessed om 22 September 2020 |
| 9 | Growth Transformation Program of Ethiopia I | http://extwprlegs1.fao.org/docs/pdf/eth144893.pdf accessed 22 September 2020 |
| 10 | Growth Transformation Program of Ethiopia II | https://www.greengrowthknowledge.org/sites/default/files/downloads/policy-database/ETHIOPIA%29%20Growth%20and%20Transformation%20Plan%20II%2%20Vol%20I.%20%20%282015%2C16-2019%2C20%29.pdf accessed on 22 September 2020 |

*2.3. Data Analysis*

The stakeholders and institutional arrangement mapping were analyzed using the engagement of different stakeholders' consultation. The collected data from key informants and the FGD were triangulated with the documents and ground observation for data validation and analysis. A document review of policies, as a major contributor to the planning and implementation of green space, was carried out. The implementation status and the challenges that hinder the direct contribution of the policies, guidelines, and regulations in the realization of green space ground implementation have been presented using the IPO (Input Process Output) model adopted from [19]. The model is employed to visualize how the inputs provided to implement green spaces undergo different processes. In the IPO model, the policy and other guiding documents were the inputs and green space development as output.

Content analysis of policy documents, guidelines, and regulations in the protection and development of green space either generally at the federal level or the city level has been made. Furthermore, principles of green space planning have been analyzed from the content of the documents. As this paper deal with the governance of green space implementation in Addis Ababa, which is the major challenge for planning, institutional arrangements in the sector of green space are mapped.

Institutional arrangements mapped by the participants involved in the FGD were based on the organogram received from the city's mayor. Triangulation was conducted for data validity, and a final organogram was developed during an FGD focusing on the line of command. Field observation and the background of current projects were analyzed qualitatively and described in the explanatory text. During ground observation, green spaces in the city such as parks, riverside vegetation, woodland area, and plantation area have been visited. Areas designated for green space development in the masterplan have been visited to triangulate data acquired from the plan commission and the masterplan of

the city. Data related to opportunities for green space development in the city and how principles of green space planning implemented on the ground have been gathered through observation at the existing green spaces. Finally, principles of urban green space planning incorporated in the 9th and 10th masterplans of the city were evaluated and compared with the green space initiatives and projects undergoing and recently completed in the city. The principles followed in the planning have been analyzed from the policy and guideline contents, which are further triangulated with ground observations of green spaces if the principles are taken into account while implementing the plan.

## 3. Results

### 3.1. Urban Green Space Policies and Implementation

Urban Planning Proclamation no 547/2008 has the scope to be applied to all urban centers throughout Ethiopia. This proclamation sought to achieve the establishment of a legal framework that promotes planned and well-developed urban centers. One plan recognized under this proclamation is the structural plan. Among the issues that the structural plan must address are principal land use classes and an environmental aspect whereby the green spaces can be addressed within these two themes in the structural plan. Furthermore, the structural plan (2017–2027) came up with a description of existing green spaces and proposes further development of the green spaces and the establishment of new parks. However, there are some urban green space projects being implemented that are not included in the 2017–2027 masterplan.

Environmental Policy of Ethiopia (1997) has a sectoral policy framework on forest, woodland, and tree resources. Under this subsection, urban forestry is overlooked, and the policy gives due emphasis on forest lands and partially on peri-urban forests. Under the subsection on human settlements, urban environment, and environmental health, the policy clearly seeks to plan and create green spaces within urban areas, including community forests and woodlands for fuel wood, recreation, providing habitats for plants and animals, as well as ameliorating urban microclimates. According to the key informants from the Addis Ababa Environment Protection and Green Development Commission, the policies are rather in the air, lacking the regulations and standards needed to enforce the policy.

Forest Development Conservation and Utilization Proclamation no 1065/2018 recognizes urban forests (roadsides, riverside, parks, and other green spaces), while the respective regulation on Forest Development Conservation and Utilization excluded the urban forestry part. To maximize the fulfillment of urban societal needs, the Ministry of Urban Development and Housing (MoUDH) prepared the Climate Change-Resilient Urban Green Development Strategy as a road map to provide urban green infrastructure.

This research used the IPO model to evaluate the significance of the policies at the federal and Addis Ababa city levels to manage the existing green spaces and establish proposed green spaces in the city. The model indicates the three processes that the policy inputs must undergo to deliver the output, the green space plan execution on the ground. In the green space development of Addis Ababa, the major inputs are supplied by the federal and city governments. Policies and planning documents prepared at both city and federal levels provide guiding principles for implementing the green space plan of the city and manage the existing green spaces. As illustrated diagrammatically in Figure 2, three processes are ongoing in order to deliver the output. These processes are:

Governance and steering process: Under this process, the actors and stakeholders who are responsible for green space planning and implementation go through task appointments and setting accountability. Though all actors have their own mandates and accountability, mandate conflict can arise due to unclear reporting lines. The planning commission, which is responsible for developing the masterplan, must work with other stakeholders and should be accountable to the city government.

Core process: The core process in the city's green space development and implementation is the development of the masterplan and other local development plans. Managing the core process plays a great role in attaining the desired output.



Auxiliary process: Under the auxiliary process, activities such as research and investigations, land use planning and consultations, as well as guiding principles produced at the federal and city levels, are managed and checked. The auxiliary process directly contributes to the core process, which is the development of the masterplan and local development plan.

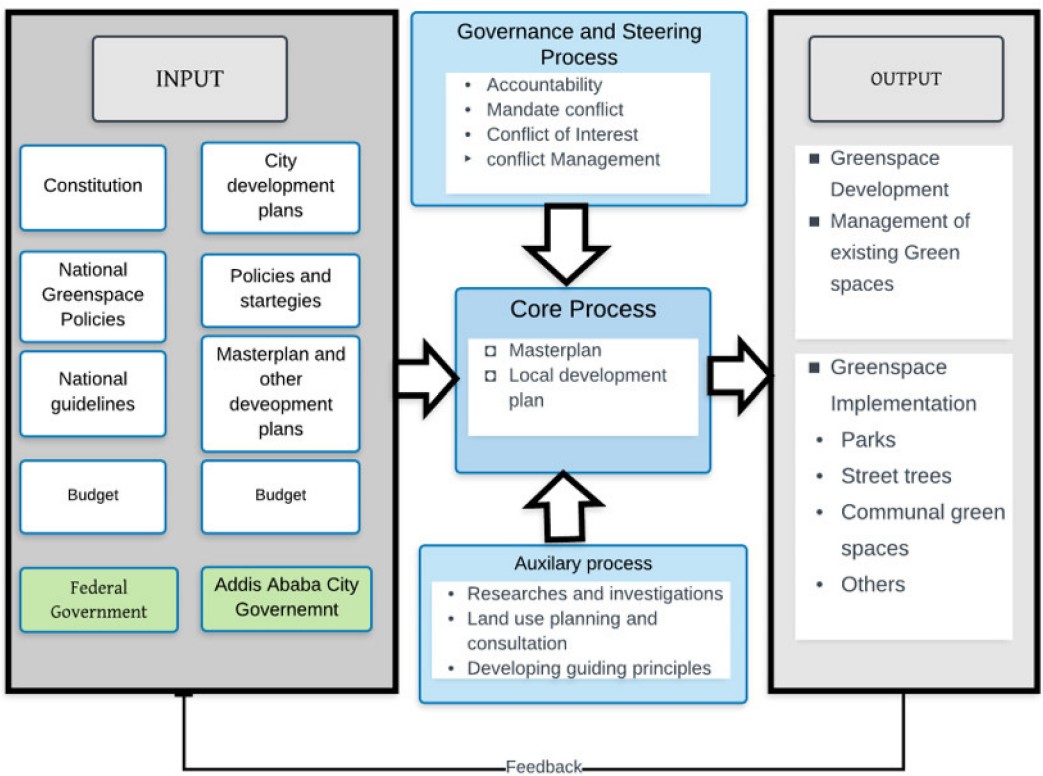

**Figure 2.** Input Process Output analysis of Addis Ababa's green space.

### 3.2. Institutional Arrangements for Green Space Planning and Implementation

The city government, including agencies that are directly involved in green space planning and implementation, is organized as shown in Figure 3. The Ethiopian Environment, Forest, and Climate Change Commission (EFCCC), a federal agency, is directly accountable to the Prime Minister, while those institutions responsible for green space planning and implementation in Addis Ababa report to the city's mayor. The three commissions selected for the key informant interview, Farmers and Urban Agriculture Commission, Addis Ababa City Environmental Protection Green Development Commission, and the City Government Plan and Development Commission, all participate in policy drafting, planning, and implementation with respect to the city's green space development. The River Basin and Green Development and Administration Agency is directly accountable to the Addis Ababa City Environmental Protection Green Development Commission, which is responsible for implementing green space plans, such as parks, riverside greening, street trees, and city squares within the city. The River Basin and Green Development and Administration Agency delegates' authority to the sub-city and woreda level for the implementation of green space development. On the other hand, the sub-cities are directly answerable to the mayor, which could make the line of communication and coordination delicate. Furthermore, linkages between the federal organization, the EFCCC, and the Addis Ababa Environment and Green Development Commission are fraught, even though both deal with environment and green spaces. Following the interviews with key informants, the organogram presented in Figure 3 was developed.

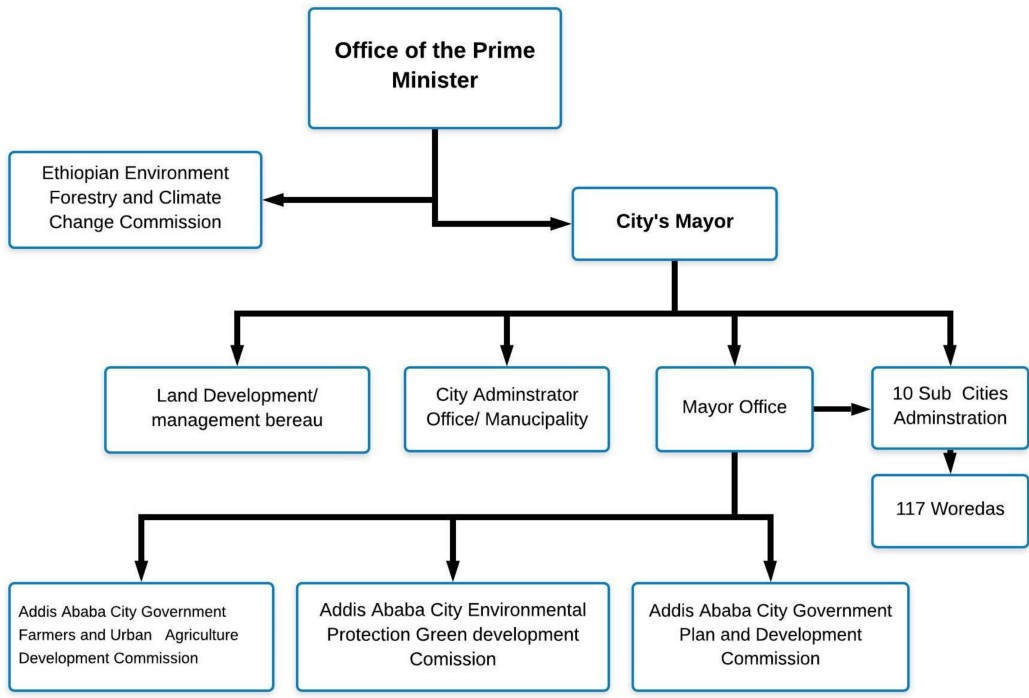

**Figure 3.** Organogram of institutions engaged in urban green space of Addis Ababa.

### 3.3. Challenges of Green Space Implementation and Positive Initiateves

#### 3.3.1. Challenges

The overall challenges to green space planning and implementation in Addis Ababa fall into four broad categories: institutional, political, social, and economic.

Institutional Challenges

According to the key informants, institutional challenges are the major challenge facing the implementation of green space policy and the planning on the ground. The inclusion of institutions in the planning process of green space plans is vital even though their direct contribution to the implementation is very low. The lack of fully engaged institutions during plan preparation is an ongoing challenge for the green space development plan and its implementation in the city. The segregated plans issued by different institutions implementing the city's masterplan fail to recognize milestones achieved by other institutions executing implementation on the same site. Institutions also fail to communicate activities realized on the ground. Miscommunication among institutions is mentioned by all key informants as a major challenge for green space development in the city. Institutions' frail communication on the activities is realized on the ground. Green spaces established on different round-about and street trees have been uprooted by the Ethiopian Road Authority. According to River Basin and Green Areas Development Agency, seven (7) roundabouts have been demolished after green area development has taken place. Additionally, Ethiopian Electric Power Corporation's action on the roadside trees is not well managed; therefore, there has been street trees removal without pre-notice to the respective organization. This shows that there is weak communication between institutions that have distinctive roles but executing their plan in the same area. Frequent institutional re-arrangement is another challenge that triggers conflict among organizations. The other challenge discussed under institutional challenge is conflicting roles between institutions.

Political Challenges

Ban forestry should be implemented based on studies and planned for the long-term. Politicizing forestry, especially in urban areas, could distort the city's plan and ruin its long-term plan. The new green legacy led Prime Minister Abiy Ahmed to initiate a campaign to plant 4 billion seedlings in 2019 across Ethiopia, including more than one

million seedlings distributed to Addis Ababa City by the River Basin and Green Areas Development Administration Agency. This specific campaign is simultaneously a positive initiative and a challenge. Urban forestry needs a long-term plan; the tree campaign rather focused on a speedy planting of trees, regardless of species or local microclimate. A well-thought-out urban forestry tree campaign needs proper planning, including the identification of species, matched to appropriate sites in advance, and the raising of seedlings in advance. Seedlings were distributed to all sub-cities, but the plantation sites were not planned in advance. Figure 4 shows that the seedlings are planted on the road median, which is not planned to be a median; it is designated as the route for an extension of the railway in the city's masterplan. It is obvious that these seedlings will be uprooted in the near future once construction commences. This shows that political engagement to receive a public hearing is a challenge for planning and implementing green spaces without contradicting with the masterplan.

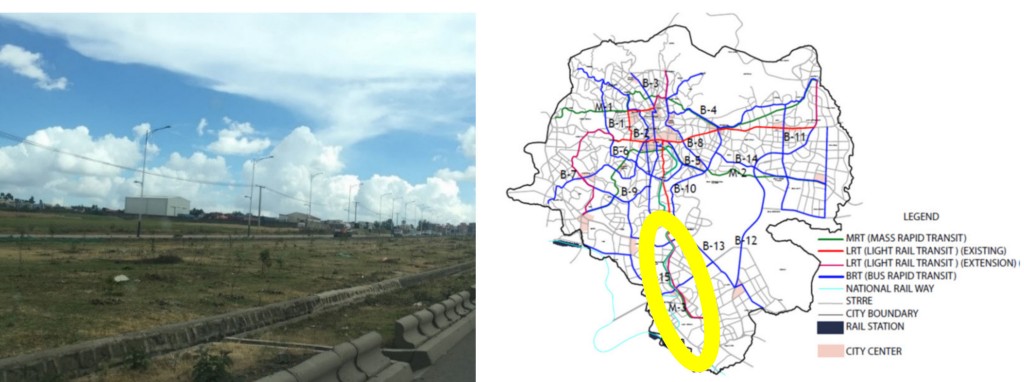

**Figure 4.** Street tree plantation made in 2019 through campaign (**left**) on area designated for railway extension on the masterplan (**right**).

Social Challenges

Society's lack of awareness of green spaces is aggravated by the ongoing rapid urbanization taking place, with it encroaching on green spaces. As shown in Figure 5, the urban population settlement critically reduces the existing green space as people move to the periphery at the expense of green areas (peri-urban forest areas). The urban population is moving toward the southern part of Addis Ababa, with agricultural fields around Kality being converted into residential areas. Development, including a highway and condominiums, is driving the expansion of the city into agricultural green space around the city.

Economic Challenges

Addis Ababa's unemployment rate is high at 23%, with 28% of the population living below the absolute poverty line. Double-digit inflation hit a high of 38% in 2018 and 2019, and life is increasingly expensive. This economic development has direct and indirect impacts on the city's green spaces. As per the discussion with the key informants, the riverside of Addis Ababa is encroached upon illegally by the poor, while river buffer development is challenged by the economic situation of city residents. Furthermore, city dwellers in Addis Ababa engage in fuelwood collection from its green spaces, a problem that is especially acute in the northern part of the city.

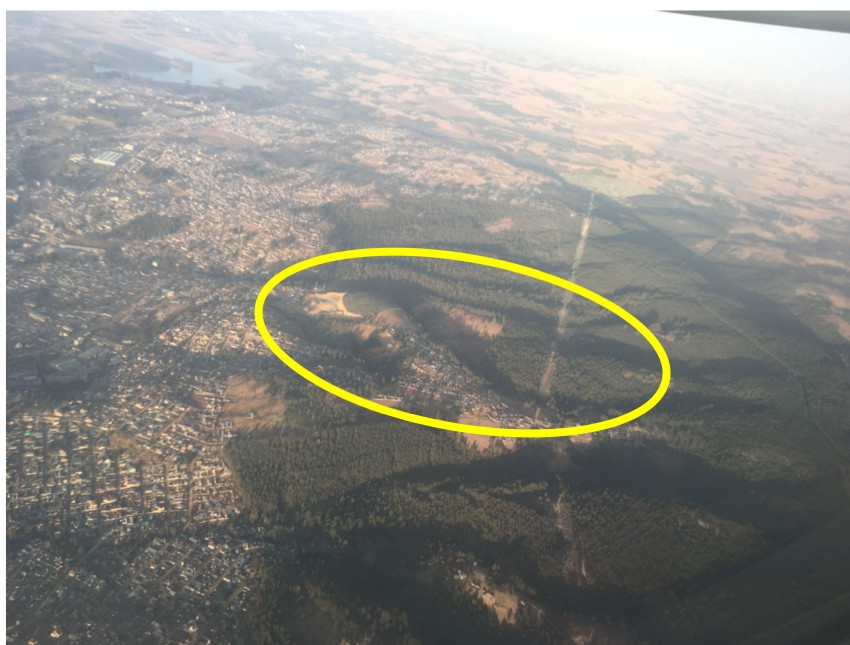

**Figure 5.** Encroachment to peri-urban forest of Addis Ababa (Photo by: Shibire Bekele, 2019).

### 3.3.2. Positive Initiatives of Greenspace Development in Addis Ababa

Political commitments

Television media coverage of Addis Ababa's green infrastructure represents a great opportunity to create awareness about the protection and management of the developed green spaces in the city. Such TV programs play a great role in the expansion of individual green areas or communal gardens in addition to managing the existing green spaces. This was noted by all key informant interviewees as a great opportunity for green space development in the city and for creating environmental awareness.

Political will and initiations by the Prime Minister and the City Mayor to work on riverside vegetation and develop green infrastructure are important. One megaproject focusing on green space development in Addis Ababa is the riverside project. The project aims to enhance the well-being of city dwellers by not just mitigating river flooding but also through the creation of public spaces and parks, bicycle paths, and walkways along the riversides. This project will run along two of the largest rivers of the city, stretching a total of 51 km, all the way from the mountain of Entoto through to Akaki River.

New technologies in urban greening

New technologies introduced to transplant big trees in the city is one of the biggest initiatives taken by the prime minister in addition to the aforementioned riverside development. Big, grown trees are uprooted from different parts of the country, then transplanted into newly established parks, replacing eucalyptus (Figure 6).

Potential of Urban and Peri-urban agriculture

The Addis Ababa City Development Plan (2002–2012) indicates that the livelihoods of 51,000 families in Addis Ababa were associated with farming. It argued that urban and peri-urban agriculture (UPA) should be encouraged, especially in the south-eastern part of Addis Ababa. It proposed horticulture development along riverbanks and livestock production on the peripheries. Overall, 13.82% of the city's land area (7175 ha) was delineated as agricultural land. However, other than this reference, UPA was not featured in the planning tools for regulating the actual implementation of the plan (i.e., Strategic Development Framework, Strategic Development Action Plans, and Local Development Plans). Existing urban agriculture and its respective area is indicated by the following pie-chart (Figure 7). As illustrated below, Akaki kaliti sub-city has the largest share of urban green space, while sub-cities, such as Addis Ketema, Lideta, Arada, Gulelle, and Kirkos, provide an insignificant amount of urban agriculture.

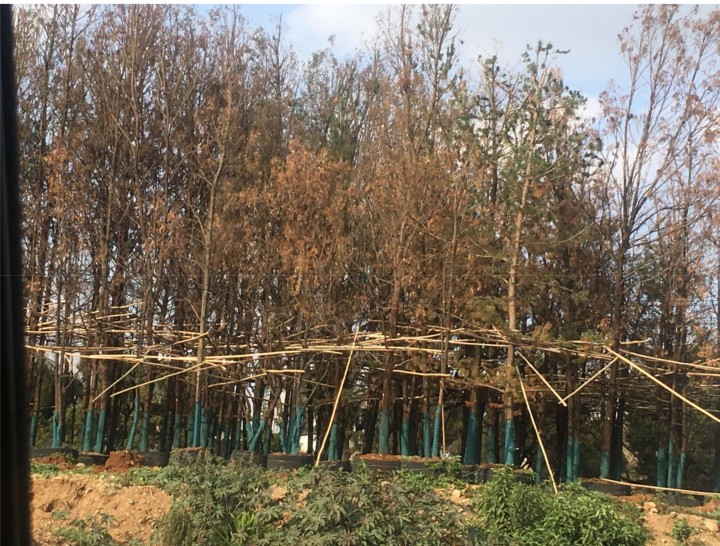

**Figure 6.** Trees uprooted from different parts of the country and collected for plantations in Addis Ababa (Photo by: Shibire Bekele, 2019).

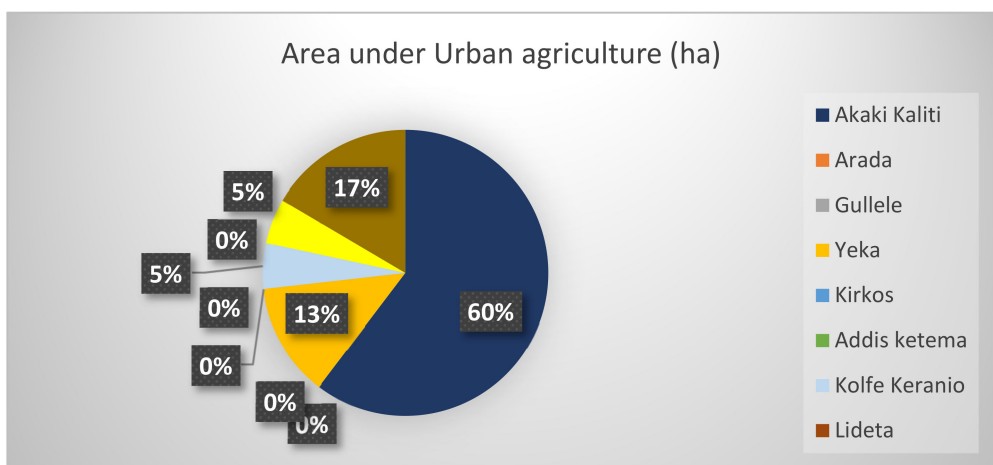

**Figure 7.** Urban agriculture area coverage at sub-city level.

Existing Greenspaces and Their Recreational Potential

There are green spaces in the city that can be used as a base to build more expansive green spaces. There are also initiatives at the local administrative level, woreda, and local community level that take a leading role in protecting open areas and turning abandoned lands into green areas. The outskirts of Addis Ababa are well covered by vegetation on mount Entoto and the Gullele Botanical Garden. Furthermore, the available green space is remnant woodlands. Building on the existing large green spaces and small patches of green spaces across the city will enhance the availability, accessibility, and use of the green spaces in the city.

*3.4. Principles Integrated in Greenspace Planning and Implementation of the Principles*

As per the discussion with key informants, different principles in green space planning have been considered during the planning phase. Multiple benefits and functions provided by different green spaces have been taken into account during the preparation of different planning documents, which are related to green spaces. The following documents integrated the green space planning principles. Through these documents, the principles of multi-functionality, green-grey integration, connectivity, and social-inclusiveness are explored, both explicitly and implicitly, in the policies and regulations.

1. The Environmental Policy of Ethiopia, which was formulated in 1997, explicitly particularized the objective to plan and create green spaces within urban areas that provide recreational activities, habitats for plants and animals and ameliorate urban microclimates.

2. The Urban Greenery and Beautification strategy, which was formulated in 2015, has the objective of developing green spaces that not just reduce environmental degradation, pollution, and urban floods but also promote environmental sustainability in the urban area.

3. The Ethiopian National Urban Green Infrastructure Standard, which was formulated in 2015, has the objective of creating ecologically well-functioning, aesthetically pleasing, and socially beneficial green spaces in cities that also provide suitable, sufficient, and ecologically viable green spaces for the recreational, social, economic, and environmental needs of the community.

4. The Green Infrastructure Based Landscape Design Supporting Manual, which was developed in 2011, also proposes developing street tree plantings for shade provision, mitigating the urban heat island effect, reducing runoff, and sequestering carbon.

From the above-mentioned policies and regulations, the principle of multi-functionality is implicitly mentioned under (i, ii and iii), while the Ethiopian National Urban Green Infrastructure Standard, formulated in 2015 (iii), explicitly indicates the principle of social-inclusiveness. The Green Infrastructure Based Landscape Design Supporting Manual (iv) implicitly indicates the principle of connectivity and green-grey integration.

## 4. Discussion

The National Urban Green Infrastructure standard of Ethiopia proposes 15 m$^2$ per capita public green open spaces within city boundaries, with every resident living within 500 m from a public green open space that is at least 0.3 hectares in size. However, the proposed standard has never been implemented accordingly in the city. Using content analysis [20,21], data from documents were analyzed, specifically identifying explicit references to various terms of green space planning principles and implicit references to related concepts like accessibility, ecological, social, and economic functions.

This study reveals that the policies are only on paper; the formulation of regulations and standards required to implement the policies on the ground is missing. The organizational structure lacks some of the key actors required to implement green space plans. The Ethiopian Environment, Forestry, and Climate Change Commission has established an urban forestry directorate. This implies that the EFCCC is engaging in urban forestry, although its link to a similar city governmental institution is not clearly defined. Among the challenges mentioned, the fast-growing urbanization, failure to implement structural plans according to planning, and weak institutional harmonization are mentioned by key informants from different institutions. Scholars also criticize inappropriate political interventions and the lack of professionals in the sector [16].

In 2014, Addis Ababa's green space was reduced to 9835 ha due to land conversion to residential (condominiums), manufacturing, and storage land uses. In 2018, 5476 ha of land was converted from green space—specifically cropland into other land uses [17]. Similarly, it is clear that the proportion of green areas has decreased, with the large-scale elimination of green spaces, including by the public sector [16,22]. As the riverside buffer vegetation suggested by the 9th masterplan was never implemented, the 10th masterplan proposes implementing the riverside vegetation. The ongoing grand Sheger project of riverside greening is part of the 10th masterplan. In the time plan of the 9th masterplan, proposals such as the riverside development were not implemented, new parks were not successfully developed, and urban agriculture was practically nonexistent [8,16]. Additionally, a buffer zone of 30 m along each riverbank should secure recreational areas for residents who live nearby [8]. This also addresses multi-functionality and the social inclusiveness principle, as most riverside residents can easily access the recreational parks along the rivers. Studies

show that social inclusiveness enhances effective management of green infrastructure and promotes community stewardship for the green infrastructure components [11,23].

## 5. Conclusions and Recommendation

Policies regarding urban green space development are well articulated, having a strong base in the constitution. Climate Resilient Green Economy (CRGE), Growth and Transformation Program 2010–2015 (GTPI), and Growth and Transformation Program 2016–2020 (GTPII) are believed to be among those strategies contributing to green space development of Addis A-baba nonetheless. These strategies focus on general forest development and conservation, lacking a clear concept of urban forest development and implementation. Regulations and directives drafted by responsible bodies are not clear, while law enforcement is hardly applied to green spaces. The organizational structure of city and federal governments has some overlapping duties that can create delays in implementation while also failing to provide proper monitoring and evaluation. There is a lack of proper implementation of the masterplan and poor communication between different sectoral organizations in planning and implementation. Furthermore, frequent organizational restructuring and reshuffling is a major challenge to the implementation of planned green space projects and keeping track of relevant activities. In the 9th Addis Ababa masterplan, 2002–2012, green space planning does not follow any planning principles. It is an important step forward that the 10th masterplan incorporates green space planning principles such as multi-functionality, connectivity, and inclusiveness.

To move green space planning and implementation in Addis Ababa forward, the research recommends establishing a platform for discussion, including annual or biannual discussions on the implementation of the masterplan. The city should create systems and forums for public participation in the decision-making process of 'key' issues. Both a national planning council and planning commission should be established to oversee the plans, based on quarterly reports, to ensure the green spaces promised in the 10th masterplan are delivered. Furthermore, monitoring and evaluation of projects and activities must be jointly carried out by involved institutions and bodies. Institutional relationships among policymaking, planning, implementation, monitoring, and evaluation bodies of the city government must be defined. Accountability loops that involve policymakers, planners, decision makers, and law enforcement at different levels from woreda, sub-city, and city government levels are critical. This will support the sharing of responsibility for project execution among engaged stakeholders, thus promoting transparency and opening the door for the wider public to be a part of the implementation process.

**Author Contributions:** S.B.E. drafted the manuscript and prepared the figures. S.B.E. and K.Y. conceptualized the research and design methodology. K.Y. and S.S. revised the manuscript. K.Y. Supervised. S.S. reviewed and edited the manuscript. All authors have read and agreed to the published version of the manuscript.

**Funding:** The research is funded by GIZ Biodiversity and Forestry program and co-financed by Addis Ababa University. The publication of the article was funded by the Open Access Fund of the Leibniz Association.

**Institutional Review Board Statement:** Not Applicable.

**Informed Consent Statement:** Informed consent was obtained from all subjects involved in the study.

**Data Availability Statement:** The dataset(s) supporting the conclusions of this article is (are) available on Google Scholar, JSTOR, Google, and all publicly available datasets are fully referenced in the reference list.

**Conflicts of Interest:** The authors declare no conflict of interest.

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
