# Peer review of "Urban Green Space Planning, Policy Implementation, and Challenges: The Case of Addis Ababa"

_sustainability, doi:10.3390/su132011344_

Round 1
Reviewer 1 Report
This paper reported the case study of Addis Abada, the capital city of Ethiopia, to address the topic of urban greening from the point of view of green space planning, policymaking, and challenges associated with public institutions' role. The topic of the paper is of key importance for the development of worldwide urban sustainability and the purposes of this paper are of primary interest for the scientific community and institutions. However, the methodology of the analysis is very poorly described raising questions on the scientific value of the investigation and the validity/significance of the obtained results. In particular, the different data sources are mostly not introduced nor described at all, and the methods the authors used to interpret the data and extrapolate a result from it. The missing information let the results and discussion sections hard to understand and so revise.
Lines 94-96: The authors should better specify and argue about the questions of the interview because they are a key data source for the analysis. What are the main question topics? How many questions have been asked? Are those open questions or yes-and-no types of questions? If open, how are the information gained analyzed and compared to each other?
Lines 106-108: What is the total number of key informants participating in the interviews? Are these same people involved in the group discussion? The number of key informants and people participating in the group is the same?
Sect. 2.2: A lot of information on the data are missing. The authors discussed only the key informant interviews and the focus group, but not the remaining data sources which remain listed but undiscussed. What type of field observation has been performed? How? For how long or what is the sampling pool? Similar questions arise for the secondary data. I think this should be integrated with the manuscript.
Lines 120-122: The triangulation process is not clear. How are the different information integrated into one another? Part of the data involved in the triangulation is not sufficiently introduced: what are the documents? What type of ground observation data are you dealing with? How do you manage these amounts of data and how do you compare the different information? What is the product of triangulation?
Line 123: The model should be discussed in more detail. How does it operate? What does it do? How does it manage the inputs to provide the outputs? What does the Process Phase do?
Lines 127-128: It is not clear to me how the input and outputs of the model relate to the collected data? How are the interviews involved in it? What about the other primary and secondary data?
Other minor comments:
Line 16: “… secondary data of review literature review …” do you mean “secondary data from a literature review”?
Line 32: Reference [33] should be [3].
Line 35: Reference here should probably be written [3, 4].
Line 44: Please correct the reference. Please check that text references and bibliography is correctly matched through the manuscript.
Lines 46-48: Here the specific case of Addis Abada is called out but not really introduced. What I want to say is that until line 46 you were giving a general introduction on the criticalities and challenges related to the topic of the paper and then you go to the specific case of the city without introducing that this will be your case study and why. For example, why choosing this city among the others? Why focusing on a single city? I think this information is somehow implicit in the current introduction, but it would be more appropriate to make it explicit to strengthen the study motivation.
Lines 48-50: what do you mean by “… the city contributes its share …”?
Lines 69-70: I suggest reporting the precise reference coordinate (in latitude and longitude) of the city and its planar extension rather than the approximate location. Also, I think the location the authors wanted to report is between 8° and 9°, so the degree symbol must be correct.
Line 70-71: by “m.a.s.l.” do you mean m a.s.l., i.e., meters a.s.l. (above sea level)? Please correct.
Line 72: “km2” the 2 should be a superscript.
Lines 72-73: What do you mean by “… land locked country.”?
Line 76: Reference error. Please correct them throughout the paper.
Lines 80-81: From the subsequent description it seems that this sentence represents the past of Addis Abada (if I understood correctly the nowadays landscape is different). If so, the use of present tense is misleading.
Line 94: What do you mean by structured and semi-structured questions? What is the difference?
Line 113: What does “FGD” stand for?
Line 116: table 2 is not mentioned in the text.
Author Response
Thank you for your valuable comments.
Please find attached our reaction to the comments provided.
Kind regards

Reviewer 2 Report
This paper intends to assess the policy and planning of urban greenspaces with their potential implementation status and challenges in planning and implementation.
- Line 44, change [109] to [10]
- Enrich the content of the introduction and increase the references cited in the introduction.
- Line 69, the Letter ‘o’ should be superscript
- Line 72, the ‘2’ of km2 should be superscript. For similar situations, please check the full text carefully
- Line 76, ‘Error! Reference source not found. shows location of Addis Ababa within Ethiopia.’ What do you want to express?
- The links of various web sites in Table 2 are best placed in references. The website should be marked with the date of query
- Line 124, [17] doesn't need to be bold.
Author Response
Thank you for the valuable comments. please find the attachment.
We have addressed the comments provided.
Kind regards

Reviewer 3 Report
The paper has a regular structure. The presented contents are clearly organized. However, the mentioned aims can be reduced (they are a little bit repetitive; if authors incorporate one or two within the remaining goals it would be more evident the main purpose of the paper). The way the research is explained is quite descriptive, lacking a more critical perspective over the research findings in order to robust the obtained results. In addition, it is not clear the sort of correlations between the green spaces of the city and its urban structure authors can advance in order to increase the role green spaces can play in the planning process of the city. This is a topic that can be better developed in the final sections of the paper. I final note the references used throughout the paper (some are missing). Finally, the scope of the paper is very relevant but authors can explore a little bit more the potential novelty of the contents and of the research outputs (again, the paper is more descriptive than critical).
Author Response
Thank you very much for the valuable comments.
The concerns are addressed in the revised version.
Round 2
Reviewer 1 Report
Key methodological information from sections 2.2 and 2.3 are still missing:
1- a clear description of the field observation: what are the collected data?
2- how are the secondary data (lines 107-108) included in the analysis?
3- my previous doubts on the triangulation have not been clarified: how are the different information integrated with one another? Part of the data involved in the triangulation is not sufficiently introduced: what are the documents? What type of ground observation data are you dealing with? How do you manage these amounts of data and how do you compare the different information?
4- Apart from the interview and FGD, it is not clear how the other data are within the model
Other minor corrections
Please correct references throughout the paper e.g., lines 46, 85-86, 203, 239, 279-280, 398
Figure 4 has two superimposed images, and the caption is separated from it
The acronyms CRGE, GTPI and GTPII are not defined (line 416)
Author Response
Thank you for the valuable comments. We have revised the article and please find attached our answer to your comments.
kind regards,

Round 3
Reviewer 1 Report
The authors made a good job answering all my comments and improving the manuscript, which can be accepted. Please, make a final check of cross-references (e.g., lines 32, 46, 244) and correct the following typos:
Lines 78-79: the geographic coordinates seem to be written with a different font name and size, please correct
Line 92: m.a.s.l. should be m a.s.l.
Line 94: "th" in 20th should probably be a superscript
Line 150: [] font should not be bold
Line 151: missing dot after “process”
Lines 155-156: “…of the documents. As this paper…” Insert space after before As
Line 158: remove the second dot
Line 182: remove space in “2017- 2027”
Line 208: “figure 2” should be Figure 2
Line 241: remove white space before “Furthermore…”
Line 330: “figure 6” should be Figure 6
Line 343: “figure 7” should be Figure 7
Line 404: remove space before “In 2008, ...” and separate the number from its unit (also at line 403, 412). I would also use square meters instead of hectares
Line 408, 441: similar to comment line 94
Line 428: remove white space before “There is a lack…”